# Breaking the Memory Barrier: Efficient Multi-Class 3D Segmentation for Hundreds of Classes

Olivier Jaubert [1]      OLIVIER.JAUBERT@MRE.MEDICAL.CANON
William Traynor [1]      WILLIAM.TRAYNOR@MRE.MEDICAL.CANON
Shadia Mikhael [1]      SHADIA.MIKHAEL@MRE.MEDICAL.CANON
John H. Hipwell [1]      JOHN.HIPWELL@MRE.MEDICAL.CANON
Sonia Dahdouh [1]      SONIA.DAHDOUH@MRE.MEDICAL.CANON

[1] *Canon Medical Research Europe, Edinburgh, United Kingdom*

**Editors:** Accepted for publication at MIDL 2026

## Abstract

Medical image segmentation has transformed clinical routine by providing fast and accurate methods for the automated measurement of biomarkers and lesions. While foundation models promise broad generalization across hundreds of anatomical structures, they often under-perform compared to task-specific deep learning methods like nnUNet. However, these specialized models face scalability challenges when segmenting large numbers of classes in 3D images. We introduce a class scalable 3D segmentation method combining a low rank basis and projection operator with a chunked cross entropy and Dice loss. This design decouples the number of classes and the peak memory requirements enabling the segmentation of hundreds of classes in 3D. Integrated into the nnUNet framework, the proposed method supports state-of-the-art training and architectures. Scalability of our framework was demonstrated by creating and obtaining high Dice scores ($> 0.95$) on a novel synthetic 3D "Toy Dataset" with up to 1000 different classes. Performance on the TotalSegmentator dataset (117 classes) was assessed showing comparable mean Dice scores between the proposed method and the multi-model TotalSegmentator baseline (0.913 vs 0.928) and outperforming VISTA3D (0.803). These results highlight a practical path toward a unified, scalable foundation model for comprehensive 3D medical image segmentation of thousands of classes.

**Keywords:** 3D medical image segmentation, nnUNet, memory-efficient, foundation models, self-supervised learning, multi-class segmentation

## 1. Introduction

In recent years, improvements in segmentation algorithms have pushed the boundaries of automated medical image analysis. Automatic measurements of biomarkers is now integrated into routine clinical analysis enabling personalized medicine which only a few years ago would have been unfeasible or too time consuming. The introduction of AI into clinical practice has sparked a surge in the development of models developed to tackle specific clinical use-cases. As an attempt to streamline these developments, image foundation models (IFM) (Kirillov et al., 2023; Ma et al., 2024; Du et al., 2023; He et al., 2024; Isensee et al., 2025; Pérez-García et al., 2025) have recently been proposed. They promise faster development and broad generalization across different tasks such as classification, report generation

and segmentation. Although foundation segmentation models show the potential to segment hundreds or thousands of different structures, their performance often falls short compared to dedicated models, such as more established U-Net architecture variants (Ronneberger et al., 2015; Milletari et al., 2016; Isensee et al., 2018). Interestingly, some models seem to straddle the space between broad foundation models and use-case dedicated architectures allowing for the fine segmentation of numerous anatomical structures while lacking the flexibility of IFMs. TotalSegmentator (TS) (Wasserthal et al., 2023) for example, harnesses the capabilities of nnUNet (Isensee et al., 2018), to segment hundreds of anatomical structures, while maintaining state-of-the-art (SOTA) segmentation performance.

3D segmentation of a high number of structures on a large image volume can be very computationally and memory intensive (especially at high resolutions). To mitigate the memory issue linked to large image size, TS has proposed running a patch-based training and inference method. However, scaling to hundreds of classes still significantly stresses GPU memory and reducing patch size degrades performance. Therefore, they divide their segmentation problem into multiple tasks ($\leq 26$ classes) with a separate model per task. While this provides high performance, scaling to thousands of classes would imply the need for many more models and would not capture inter-class relationships between models. A key issue is that, traditionally, the loss computation requires the materialization of the full sized logits which increases linearly with the number of classes.

In the Natural Language Processing (NLP) community a similar issue was encountered when using a very large sized dictionary as vocabulary. In their case, the cross-entropy loss has a memory footprint which grows with the product of vocabulary size and number of tokens per batch and is responsible for up to 90% of the memory footprint of modern Large Language Model (LLM) training (Wijmans et al., 2024). Multiple methods were proposed to deal with the issue to reduce memory by avoiding full logits calculation (Milakov and Gimelshein, 2018; Hsu et al., 2024; Wijmans et al., 2024). However, to the best of our knowledge, these have not been investigated for 3D segmentation tasks where the memory footprint of categorical cross-entropy and Dice losses scales with batch size, spatial size and number of classes.

Current foundation models propose the handling of large sets of classes by using binary cross entropy separately for each class, treating the problem as a binary classification problem for each input prompt (Ma et al., 2024; Du et al., 2023; He et al., 2024; Butoi et al., 2023). While this solves the issue of memory requirements for the loss, it leads to independent treatment of labels. Conversely, using categorical cross entropy solves a true multi-class problem capturing inter-class relationships.

Inspired by recent LLM work (Wijmans et al., 2024) that aimed at reducing memory, we propose a class scalable method for 3D segmentation able to segment hundreds of classes while keeping a high segmentation performance and a manageable memory footprint. Our contributions are threefold. First, we propose a novel class scalable 3D segmentation method featuring i) a low rank basis feature space, ii) a linear projection operator that maps the basis features to logits and iii) a chunked Cross entropy + Dice loss that never materializes the full logits. Second, we have built a novel Toy Dataset to demonstrate the scalability of the proposed architecture on 3D segmentation tasks of up to a thousand classes. Finally, the new segmentation head and loss are integrated into the nnUNet framework which allows

for SOTA training and evaluation. A comprehensive evaluation of the method is performed on the proposed Toy dataset and on the TS CT dataset trained on 117 targets.

## 2. Methods

### 2.1. Architecture: General Overview

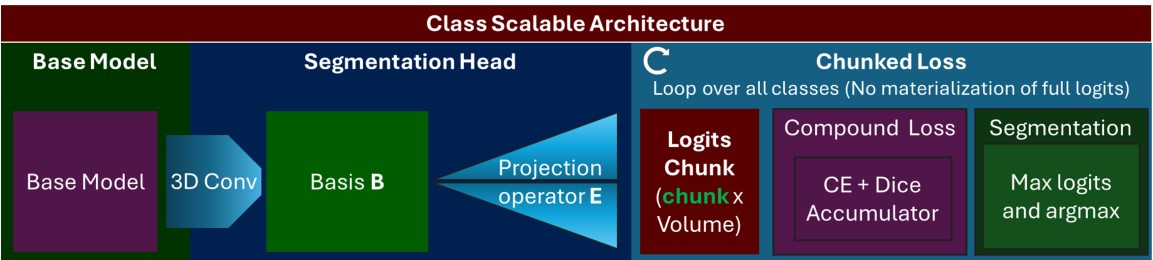

Figure 1: **Class Scalable Segmentation Architecture.** A base model is linked to a segmentation head allowing to create logits for subsets of classes. The loss and segmentation maps are obtained by looping over chunks of classes.

3D medical image segmentation presents unique computational challenges due to the higher dimensionality of volumetric data and the potentially large number of anatomical or pathological classes involved. A major bottleneck arises during training when computing the categorical cross-entropy loss over large volumes and extensive class sets. Traditional approaches require materializing the full class-volume tensor of logits which scales poorly with both volume size and class count. This leads to excessive memory consumption, often exceeding the capacity of modern GPUs and limiting batch size, patch size, or model complexity.

Figure 1 presents a general overview of the proposed method in which we decouple the number of classes and the memory footprint for memory-efficient training and inference. The proposed framework is comprised of the following elements:

1. **A Base Model**: any nnUNet compatible model - we use the nnUNet framework to harness its architectures, data pre-processing, data augmentation and training and inference pipelines.

2. **A Segmentation Head**: composed of i) a 3D Pointwise (kernel size of 1) Convolution that produces a fixed set of $R$ basis feature maps $B$ and ii) a linear operator $E$ that maps the basis features to logits for any class. This allows us to compute logits selectively without materializing the full class-volume tensor.

3. **A Chunked Loss**: The forward and backward passes of the compound (Dice and Cross entropy) loss used in nnUNet are adapted to be computed from the basis and E operator by iterating over all classes.

## 2.2. Architecture: Class Scalable Segmentation Head

Traditionally, segmentation networks map the input $x$ to outputs of size $b \times C \times S$ (i.e the full class-volume tensor) where $b$ is the batch size, $C$ represents the number of output classes and $S$ the flattened 3D spatial size. We propose to replace the final layer of the base model with a pointwise 3D convolution, which maps to $R$ basis feature maps $B \in \mathbb{R}^{b \times R \times S}$, and use a simple linear operator $E \in \mathbb{R}^{R \times C}$ for class projection. This enables the computation of logits in chunks of classes decoupling the number of classes from the output size and peak memory required. Coupled with a custom chunked loss as described in 2.3, the networks with the proposed head can be trained on the segmentation of a large number of classes without ever materializing the full class-volume tensor.

Similarly at inference, to avoid prohibitively large matrices, segmentation maps can be retrieved by looping over all classes and updating the index and value of the maximum logits observed in each voxel to obtain the final integer label map.

Additionally, nnUNet uses a sliding window inference scheme where overlapping patches are merged by averaging the logits just before final class prediction. With a large number of classes, this scheme leads to the storing of very large matrices. In order to never materialize the full class-volume tensor for thousands of classes over a large field of view, we average the basis maps over the overlapping regions before projection, allowing us to, once again, decouple the number of classes from the output size.

We evaluated the framework using the ResEnc-L (Isensee and Maier-Hein, 2019; Isensee et al., 2024) base model on which we attached the proposed segmentation head.

## 2.3. Architecture: Chunked Loss

As the logits never fully materialize, the traditional cross-entropy and Dice loss had to be modified into a chunked loss as follows:

- **Inputs** The proposed loss takes the basis feature maps $B \in \mathbb{R}^{b \times R \times S}$ and $E \in \mathbb{R}^{R \times C}$ operator as input as well as the single channel integer targets $y \in \{0, \ldots, C-1\}^{b \times S}$.

- **Forward computation** To never fully materialize the full logits, the forward pass was re-implemented to stream computations over $N_{chk}$ chunks of size $S_{chk}$ of data. This is performed by a first pass chunkwise computation of the numerically stable max-substracted log-sum-exp (Wijmans et al., 2024) over all classes $c$ yielding the denominator of the categorical cross-entropy: $\text{denom}_{b,S} = \log \sum_c \exp(\ell_{b,c,S})$ where $\ell_{b,c,S}$ represents the output logits. A second streaming is then performed to obtain softmax probabilities $p_{b,c,S} = \exp(\ell_{b,c,S} - \text{denom}_{b,S})$ for cross entropy and soft Dice calculation.

- **Loss** The losses are accumulated over all class chunks and defined as

  (1) $L_{\text{CE}} = -\sum_{n=1}^{N_{chk}} \sum_{c=1}^{S_{chk}} y_{b,c,S} \cdot \log(p_{b,c,S})$

  (2) $L_{\text{Dice}} = -\frac{1}{N_{chk} \cdot S_{chk}} \sum_{n=1}^{N_{chk}} \sum_{c=1}^{S_{chk}} \left( \frac{2 \sum_S (p_{b,c,S} \cdot y_{b,c,S}) + \texttt{smooth}}{\sum_S y_{b,c,S} + \sum_S p_{b,c,S} + \texttt{smooth}} \right)$

  The final loss is a weighted sum of the two latter:

  (3) $L = weight\_ce\, L_{\text{CE}} + weight\_dice\, L_{\text{Dice}}$.

- **Backward computation** The backward pass, required to be streamed over chunks as well, to avoid generating large matrices during gradient calculation. To reduce computation overheads, the denominator of the cross-entropy ($\text{denom}_{b,s}$) and Dice statistics ($\text{inter}_c, \text{sum\_true}_c, \text{sum\_prob}_c$) are stored during the forward pass.

---

**Algorithm 1:** Pseudo code for chunked loss calculation

---

**Input:** basis $\in \mathbb{R}^{b \times R \times S}$, operator $E \in \mathbb{R}^{R \times C}$, labels $y \in [0, C)^{b \times S}$
**Output:** $L$, the compound loss
$\text{denom}_{b,S} \leftarrow stream\_logsumexp(basis, E)$;
$L_{\text{CE}} \leftarrow 0, \ L_{\text{Dice}} \leftarrow 0$;
**for** *class chunk* $\mathcal{C}$ **do**
$\quad z_\mathcal{C} \leftarrow \text{MATMUL}(basis, E_{[:,\mathcal{C}]})$
$\quad p_C = \exp(\ell_C - \text{denom}_{b,S})$
$\quad L_{\text{CE}} += - \sum\limits_{c \in \mathcal{C}} y_{b,c,S} \log p_{b,c,S}$
$\quad L_{\text{Dice}} += - \sum\limits_{c \in \mathcal{C}} \frac{2 \langle p_c, \mathbf{1}[y=c] \rangle + \epsilon}{\|p_c\|_1 + \|\mathbf{1}[y=c]\|_1 + \epsilon}$
**end**
**return** $weight\_ce \ L_{\text{CE}} + weight\_dice \ L_{\text{Dice}}$

---

Pseudo code for chunked loss calculation is provided in Algorithm 1. By cycling over all classes, the proposed loss efficiently computes the compound cross-entropy and Dice loss and backward gradients and does so without any approximation. This enables the training of a network for the segmentation of a large number of classes.

## 2.4. Toy Dataset

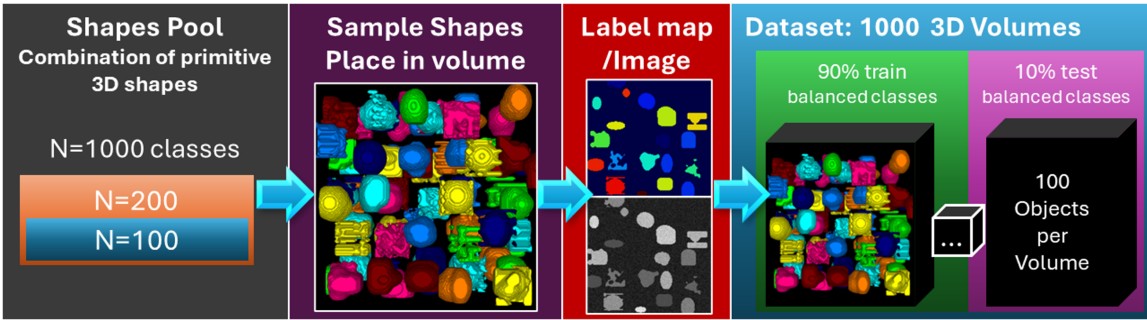

Figure 2: **Toy Dataset.** Shapes sampled from combinations of primitives are placed in a 3D volume and an image and label maps are created. This process is repeated 1000 times to create a balanced dataset with 900 training/100 test examples.

To investigate the scaling capabilities of the proposed method, a Toy Dataset is created as illustrated in Figure 2 and as described below.

- **Shape generation** We generate $C$ different shapes. Each shape is a 3D binary voxel mask within a 30x30x30 box. Shapes are created pseudo-randomly (seeded) by composing 2 to 4 random primitive shapes: Ellipsoids, Superellipsoids, and Perlin-noise blobs (thresholding of 3D noise). The union of primitives is limited by iterative erosion if the coverage of the box exceeds 60%, and is then cleaned with opening and closing operations. Each class is represented by its shape and attributed mean $\mu_c$ and standard deviation $\sigma_c$ value representing that particular class' grayscale value.

- **Image and labels creation** $N_{objects}$ are placed in a volume avoiding overlap using a greedy placement algorithm. Once the label map is created, each object pixel's value is sampled from the normal distribution with $(\mu_c, \sigma_c)$. Additional noise is added to the background.

- **Dataset Creation** The datasets are created directly in the nnUNet format with the following parameters: number of 3D volumes $N_{volumes}$, train/test split of 90%/10% volumes, volume size $V_s = (192, 192, 192)$, number of objects per volume $N_{objects}$. Each 3D Volume V contains shapes defined by an arithmetic progression modulo $C$ with $S_{(g+k*step)_{modC}}$ for $k = 0, 1, ..., N_{objects}$ with $step = C/N_{objects}$ and $g$ the volume number. This leads to well balanced train and test sets with an equivalent number of instances for each class.

While the geometrical shapes created are not diagnostically meaningful, they are distinguishable and perfectly labeled. This allows us to investigate the feasibility of our approach in terms of performance, memory consumption, training and inference times when scaling the number of classes. A total of 3 toy datasets were created : Toy 100, Toy 200 and Toy 1000 which respectively sampled 100, 200 and 1000 different classes (i.e. $C = 100, 200$ and 1000 respectively). Each dataset included $N_{volumes} = 1000$ volumes each including $N_{objects} = 100$ different objects per volume.

### 2.5. Experiments

All networks in these experiments were trained for 500 epochs (i.e. 125k iterations) using a ScheduleFree AdamW optimizer (Defazio et al., 2024) using the recommended hyper parameters winning the self tuning entry of the MLCommons algorithm benchmark (MLCommons, 2024) (i.e. $lr = 0.0025$, $1 - \beta_1 = 0.1$, $\beta_2 = 0.995$, $weight\ decay = 0.0812$ and $warmup\ factor = 0.02$). The model with the best exponential moving average validation Dice score is selected as the final model to avoid overfitting.

Loss weights $weight\_ce$ and $weight\_dice$ were both set to 1. Since a perfect Dice leads to $L_{\text{Dice}} = -1$, the minimum compound loss $L$ attainable is -1.

Each network was trained on a single NVIDIA Tesla V100 SXM2 32GB GPU. Data pre-processing included Z normalization, $1.5mm$ isotropic resampling and cropping. Data augmentation techniques are applied similarly to TS during training and included spatial (rotations, scaling and *no mirroring* to avoid left/right confusion) and image transforms (Gaussian noise, Gaussian blur, brightness, contrast and gamma correction).

Experiments are described below:

1. **Loss profiling**. The new loss was investigated in terms of memory requirements and runtime with respect to chunk size ($chunk = 15, 30, 60, 120, 1000$), number of classes ($C = 100, 200, 500, 1000$) and spatial size ($S = (32, 32, 32), (128, 128, 128), (192, 192, 192)$). Profiling was performed over 10 runs and after 3 warm-up runs.

2. **Toy Dataset Experiment**. Following nnUNet's recommendations (Isensee et al., 2024), we chose ResEnc-L ($R = 32$) as our base architecture and trained it on the Toy 100, 200 and 1000 datasets (see 2.4). Once trained, Dice score was used to evaluate the models on the respective test sets (N=100) using sliding window inference.

3. **Optimization**. The proposed method introduced two hyper parameters, chunk size and the number of feature basis maps $R$. Chunk size only affects memory and runtime making $R$ the only additional hyper parameter affecting segmentation performance to be investigated. ResEnc-L models with $R = 1, 2, 4, 8, 16$ and $32$ were trained on the TS dataset. Since one of the advantages of the proposed solution is the lower memory requirements, we increase the patch size from $S = (128, 128, 128)$ to $S = (160, 160, 160)$ and $S = (192, 192, 192)$ while still segmenting all 117 classes. Chunk size could be set at 30 for all experiments except with $S = (192, 192, 192)$ where chunks of 10 were used.

4. **TotalSegmentator Assessment**. The proposed model with parameters $R = 32$, $S = (192, 192, 192)$ and $chunk = 10$ was assessed on the TS test set in comparison to TS (v2) baseline (reported results) and VISTA3D. The test set is composed of 89 images and contains 117 classes. Images are resampled at a resolution of $1.5mm^3$ with a mean image size of $(244\pm62, 234\pm62, 287\pm136)$ pixels.

## 2.6. Metrics and Statistics

Segmentation quality was assessed using Dice scores. For comparisons, the distributions were tested for normality using a Shapiro-Wilks test. As normality could usually not be assumed, we used a non-parametric Wilcoxon signed ranked test. Statistical differences ($p < .05$ with Bonferroni correction when appropriate) and effect size (*eff_size* using the rank biserial correlation coefficient) are reported.

## 3. Results

### 3.1. Loss profiling

Results are summarized in Table 1 (Full Table in Appendix 4). Small chunk sizes led to low memory requirements while only introducing a small speed reduction compared to full logits size (chunk 1000). As per design, the peak memory usage is not dependent on the number of classes. However, runtime increases linearly with the number of classes. Finally, the method enabled loss computation on large patch sizes (192x192x192) on 1000 classes without reaching GPU memory limitations.

Table 1: **Loss profiling.** Loss only running time and peak memory usage for forward and backward passes for variable chunk size, number of classes and patch sizes (PS).

| | | Time (s) | | Memory (MB) | |
|---|---|---|---|---|---|
| | Total | Forward | Backward | Forward Peak | Total Peak |
| **Variable chunk size with Classes=1000 and PS=32** | | | | | |
| chunk 15 | 0.18 | 0.05 | 0.13 | 37.64 | 51.15 |
| chunk 30 | 0.13 | 0.03 | 0.09 | 50.33 | 74.12 |
| Direct loss (chunk 1000) | 0.11 | 0.03 | 0.08 | 843.97 | 1198.83 |
| **Variable n classes with Chunk=30 and PS=32** | | | | | |
| n classes 100 | 0.02 | 0.00 | 0.01 | 50.10 | 73.77 |
| n classes 1000 | 0.13 | 0.03 | 0.09 | 50.33 | 74.12 |
| **Variable Patch size with Chunk=10 and Classes=1000** | | | | | |
| Patch size=128 | 0.90 | 0.22 | 0.68 | 1112.41 | 1822.43 |
| Patch size=192 | 3.00 | 0.74 | 2.26 | 3715.41 | 6111.68 |

### 3.2. Toy Dataset Experiment

Table 2: **Toy Dataset.** Evaluation of Validation and Test Dice, Best Epoch for model selection and training time for the full 500 epochs.

| Dataset | Model | Val. Dice | Test Dice | Best Epoch | Training time (hours) |
|---|---|---|---|---|---|
| Toy 100 | ResEnc-L R32 | 0.998 | 0.967 | 446 | 55 |
| Toy 200 | ResEnc-L R32 | 0.998 | 0.963 | 493 | 61 |
| Toy 1000 | ResEnc-L R32 | 0.994 | 0.959 | 498 | 219 |

All models led to high Dice scores ($> 0.95$) on all three toy tasks (Table 2), even on the Toy 1000 dataset, as shown qualitatively in Appendix 5. Training on 1000 classes took approximately 4 times longer than training on 100 classes for the same number of epochs and showed slower convergence as seen in the training curves provided in Appendix 6.

### 3.3. Optimization

Figure 3 shows the first and mean of basis maps as well as output segmentations for models trained with $R = 2, 4, 8, 32$. The model $R = 1$ was stopped due to early convergence and a validation Dice of 0. Training times and validation Dice were respectively [25.1, 25.1, 32.8, 33.2, 34.6] hours and [0.161, 0.853, 0.930, 0.932, 0.932] for models with $R = 2, 4, 8, 16, 32$. Training curves showed that $R <= 4$ feature maps were insufficient while $R = 8$ feature maps already provided results close to $R = 32$ albeit slightly worse and with

slower convergence. Qualitatively, first and mean of basis maps seem different for different ranks, however they all reflect the subject anatomy and lead to similar segmentations for $R >= 8$. While $R = 16$ could be sufficient and have a lower memory footprint, since the overall aim is to encode more than 117 labels, $R = 32$ was preferred for subsequent experiments.

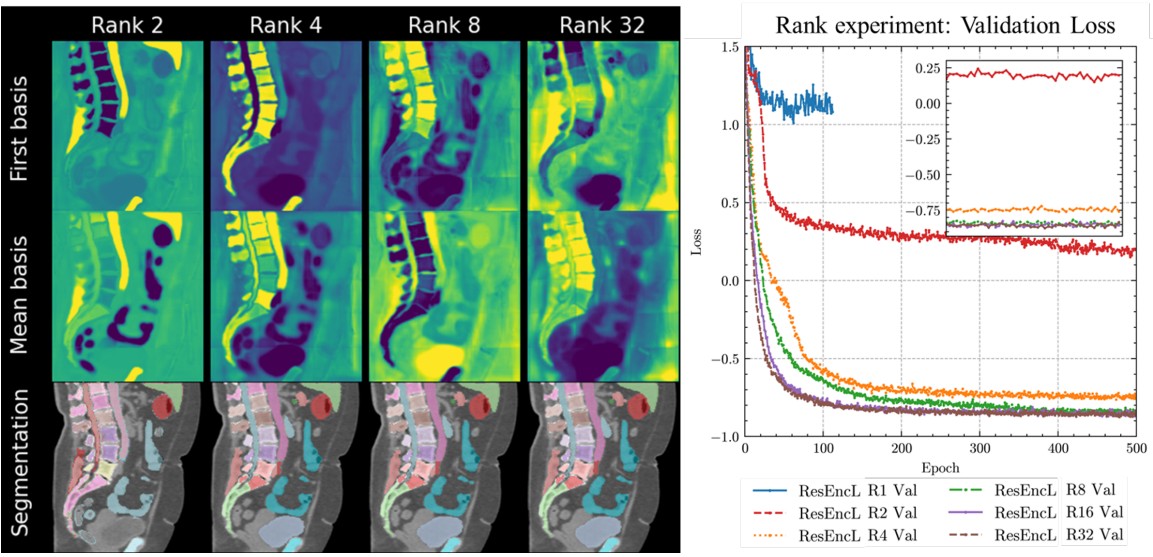

Figure 3: **Rank Experiment. Left:** First and Mean basis map and segmentation of a TS test subject for models with rank $R = 2, 4, 8, 32$. **Right:** Validation curves for $R = 1, 2, 4, 8, 16, 32$.

Models trained with patch sizes of 128, 160 and 192 and $R = 32$ led to training times of 35, 66 and 114 hours, validation Dices of 0.932, 0.941, 0.945 and test Dices of 0.903, 0.908 and 0.913 respectively. Corresponding training curves and validation Dice scores are shown in Appendix 7. A patch size of 192 led to statistically significantly higher test Dice scores compared to a patch size of 128 ($p = 7e - 15$, $eff\_size= 0.84$) and 160 ($p = 1.5e - 09$, $eff\_size= 0.64$). Mean sliding window inference times on the TS test set took 17, 14.6 and 14.4 seconds per full size image for models with patch sizes of 128, 160 and 192 respectively.

Profiling of the final proposed model showed that it's training step (batch size of 2, chunk of 10, 117 classes, patch size of 192) took 2.9s including 0.24s for forward pass, 0.78s for loss computation and 1.88s for backward pass with 29.9 GB peak memory allocated (fitting on a single 32GB GPU). While inference of a single patch with batch size of 1 took 0.27s, including 0.13s for forward pass and 0.13s for segmentation map generation with peak memory consumption of 4.7GB.

### 3.4. TotalSegmentator Assessment

The proposed model ResEnc-L, (patch size 192x192x192 and $R = 32$) is compared to TS and VISTA3D results on the TS test set. Noteworthy, the TS method was trained on a bigger

dataset than our proposed method. Results are compared qualitatively and quantitatively in Figure 4 and table 3. The proposed method achieved an overall Dice value of 0.913 which was statistically significantly higher than VISTA3D (0.803, $p < 0.001$, *eff_size*= 0.97) but slightly lower than TS (0.928, $p < 0.001$, *eff_size*= $-0.65$). A 3DSlicer (Fedorov et al., 2012) visualization of the proposed segmentation of another test subject is provided in Appendix Figure 8.

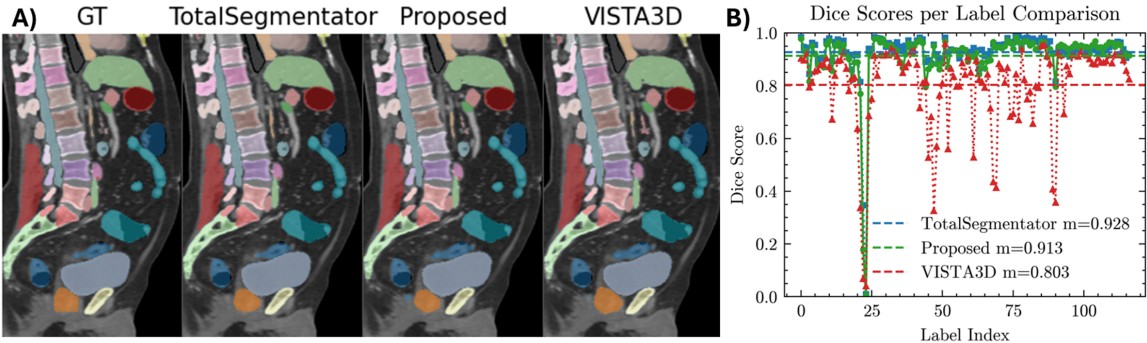

Figure 4: **TotalSegmentator Comparison.** A) Sagittal slice of a test subject comparing the proposed method, TS and VISTA3D to the Ground Truth labels. B) Test set results for the three methods showing per label and overall mean Dice score.

Table 3: **Results Table.** Mean dice scores per groupings for Bone, Muscle, Cardiovascular and Internal Organ labels and over all labels

|  | Bones | Muscles | Organs | Cardiovasc. | Overall |
|---|---|---|---|---|---|
| TotalSegmentator | 0.944 | 0.974 | 0.865 | 0.926 | 0.928 |
| Proposed | 0.937 | 0.951 | 0.838 | 0.908 | 0.913 |
| VISTA3D | 0.823 | 0.829 | 0.757 | 0.806 | 0.804 |

## 4. Discussion

The proposed class-scalable 3D segmentation method enables the segmentation of thousands of anatomical classes by breaking the memory usage barrier reached when creating the full class-volume logits tensor for loss calculation.

The chunked cross entropy and Dice loss enabled decoupling of the number of classes and peak memory usage. This truly allowed for class scalability. The tradeoff however, was longer training times when mapping to larger number of classes. It is worth noting that we did not optimize training times as it is beyond the scope of this paper. It is however a known limitation when considering scaling to a very high number of classes. Similar

tradeoffs were observed in the NLP community and solutions optimizing loss computation using information on sparsity of losses and better kernels were proposed (Wijmans et al., 2024). We believe similar solutions could be used to reduce the computational overhead.

To evaluate scalability to a thousand classes in a controlled setting, we introduced a synthetic 3D Toy Dataset. This dataset provides a valuable benchmark for testing segmentation performance at scale. Additionally, the Toy dataset has no segmentation ambiguity (perfect labeling of well defined structures) or class imbalances (instances are all equally represented at training, validation and testing) which enabled us to demonstrate feasibility and even high performance of existing networks at segmenting a thousand classes in an ideal setting.

Initial experiments on TotalSegmentator also showed the proposed method's effectiveness for 117 classes in a real world application. Not all labels were equally well segmented and Dice scores were overall lower than on the Toy datasets, reflecting the added complexity of the real world application such as large class imbalances, imperfect labeling and ambiguity in structures. Results were close, albeit slightly inferior (-0.015 Dice points), to the TS multi-model baseline. Differences were expected as TS was trained on a larger dataset than the one publicly available and the training is non-deterministic.

VISTA3D also proposed a class scalable approach using binary cross entropy for each label separately. The model possesses an automatic and point promptable branch and was trained on multiple datasets (127 total classes), including TS, allowing us to assess the performance of its automatic branch. The authors report that training VISTA3D required 64 32GB NVIDIA V100 GPUs with around 20,000 total GPU hours which is orders of magnitude superior to the proposed model (114 hours on a single 32GB NVIDIA V100 GPU). Despite its currently narrower scope and smaller training dataset, our method proposes a simpler approach with a smaller gap in performance with the SOTA and orders of magnitude lower GPU requirements. Swin SMT (Płotka et al., 2024), a recently proposed multi-class segmentation method trained and tested on the same TS dataset as in this paper reports using a DGX workstation equipped with 8 NVIDIA A100 40GB GPUs for training their method. Our method presents a more scalable approach with better overall performance (0.91 vs 0.85 for Swin SMT) and lesser computational requirements. Interestingly, they present a benchmark comparison on which their proposed method outperformed several state-of-the art segmentation methods on the TS dataset.

The proposed single unified model reduces training and inference complexity when compared to TS or VISTA3D frameworks while enabling scalability with minimal loss in performance. There is a latency penalty compared to standard nnUNet inference due to the extra calculation of the basis and chunking of the map generation. However, since it enables segmentation by a single model rather than multiple models, it becomes more efficient when many classes are involved. For instance, for the TS task, five separate models with the equivalent backbone, applied sequentially, would lead to inference times of 0.67s for a single patch while the proposed approach requires 0.27s. The additional hyperparameter $R$ was investigated and a value of $R = 8$ shown to lead to good performance on the 117 labels TS task. This could help reduce peak memory requirements drastically when critical. However, in our application, $R = 32$ was preferred as the GPU memory allowed this and it led to only a small training time overhead, slightly higher validation Dice and faster convergence. The aim was to avoid this particular layer becoming a performance bottleneck, especially

when scaling the problem to more than 117 labels. While R=32 led to good performance for both the 117 anatomical classes of TS and the 1000 synthetic classes of the Toy Dataset, questions remain on the ability of the low-rank basis to encode very high numbers of classes presenting different degrees of spatial coherence and sparsity.

We demonstrated our method on an ideal Toy dataset with 1000 classes and on a well curated public dataset (TS) with 117 classes. Further investigation is needed to validate the proposed method on a wide collection of heterogeneous (modality: CT, MR, PET or applications: oncology, brain, etc) real life datasets similarly to the recent Segment Anything with Text (SAT) (Zhao et al., 2025) models (497 target classes). However, we expect that handling class imbalances and sparsity will be necessary to truly scale to multiple heterogeneous datasets and thousands of classes. Careful attention to sampling during training and the use of cost sensitive learning as well as other strategies(Salmi et al., 2024; Chawla et al., 2002) will be investigated.

## 5. Conclusion

In conclusion, a class-scalable 3D segmentation method combining chunked loss and segmentation head enabling segmentation of potentially thousands of classes using a single model was proposed and demonstrated on synthetically created data overcoming key memory limitations of previous approaches. Our integration with nnUNet enabled high-performance segmentation as demonstrated on the TotalSegmentator task. The simplicity and scalability of our approach offer a compelling alternative for unified segmentation, breaking the previous memory barrier experienced when computing the categorical cross entropy loss on large numbers of anatomical classes. Future work will focus on refining the loss function and using a collection of diverse and relevant datasets to increase the number of available segmentation classes.

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

# 6. Appendix

## 6.1. Loss Profiling

Table 4: **Profiling of the loss.** Full Table. Loss running time and peak memory usage for forward and total (forward and backward passes) for variable chunk size (with 1000 classes and Patch Size (PS) 32), variable number of classes (with Chunk=30, PS=32) and variable patch sizes (with classes=1000 and chunk=10)

|  | Time (s) | | | Memory (MB) | |
|---|---|---|---|---|---|
|  | Total | Forward | Backward | Forward Peak | Total Peak |
| **Variable chunk size with Classes=1000 and PS=32** | | | | | |
| chunk 15 | 0.18 | 0.05 | 0.13 | 37.64 | 51.15 |
| chunk 30 | 0.13 | 0.03 | 0.09 | 50.33 | 74.12 |
| chunk 60 | 0.11 | 0.03 | 0.08 | 74.20 | 114.31 |
| chunk 120 | 0.11 | 0.03 | 0.08 | 128.96 | 205.33 |
| chunk 1000 (full logits) | 0.11 | 0.03 | 0.08 | 843.97 | 1198.83 |
| **Variable n classes with Chunk=30 and PS=32** | | | | | |
| n classes 100 | 0.02 | 0.00 | 0.01 | 50.10 | 73.77 |
| n classes 200 | 0.03 | 0.01 | 0.02 | 50.12 | 73.81 |
| n classes 500 | 0.06 | 0.02 | 0.05 | 50.20 | 73.93 |
| n classes 1000 | 0.13 | 0.03 | 0.09 | 50.33 | 74.12 |
| **Variable Patch size with Chunk=10 and Classes=1000** | | | | | |
| Patch size=32 | 0.03 | 0.01 | 0.02 | 33.66 | 45.02 |
| Patch size=128 | 0.90 | 0.22 | 0.68 | 1112.41 | 1822.43 |
| Patch size=192 | 3.00 | 0.74 | 2.26 | 3715.41 | 6111.68 |

## 6.2. Toy Dataset Experiment

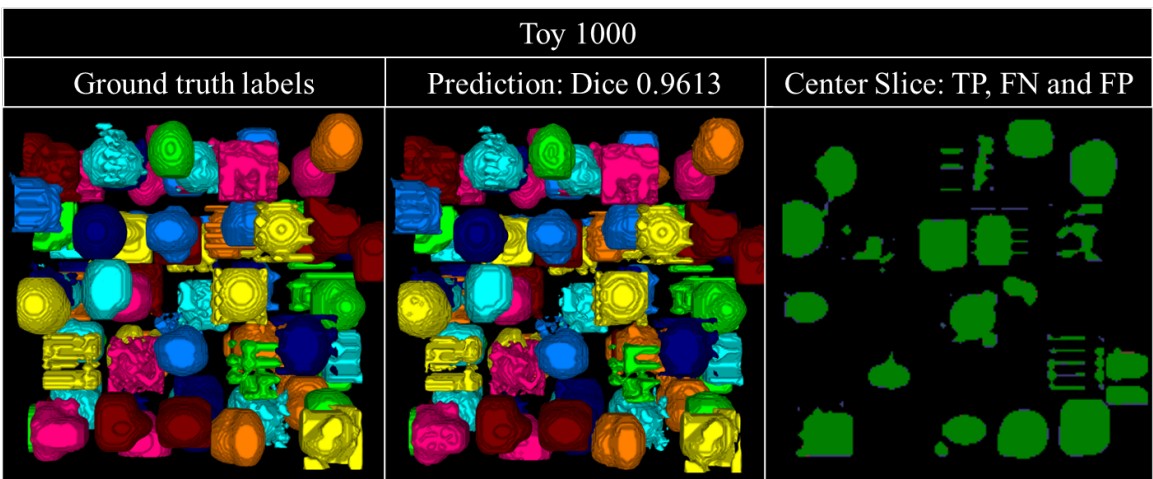

Figure 5: **Qualitative example of Toy 1000 dataset.** Comparison of 3D visualizations of Ground truth and Prediction, and for one slice True Positives (TP in green), False Negatives (FN in blue) and False Positive (FP in red).

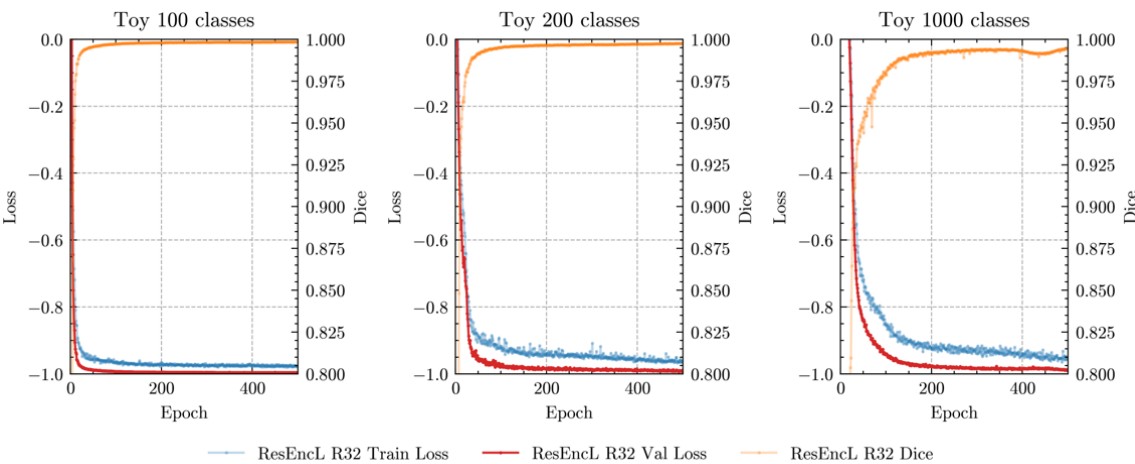

Figure 6: **Toy Dataset Experiment.** Training loss, Validation loss and Validation Dice scores for Toy 100, Toy 200 and Toy 1000 datasets

## 6.3. Optimization

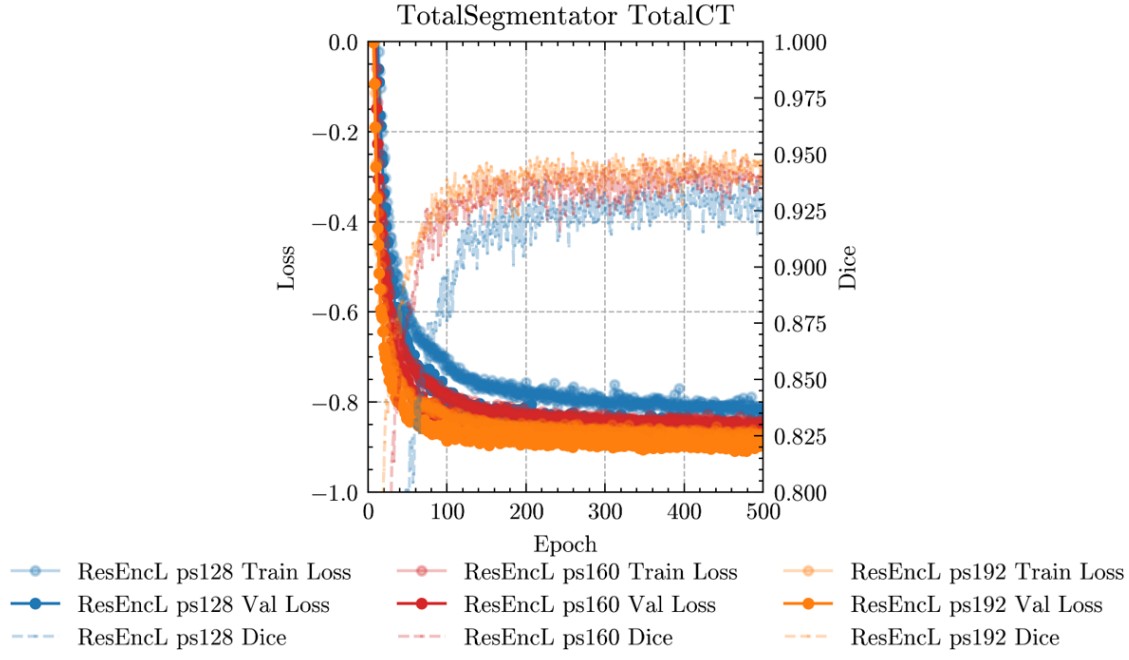

Figure 7: **Patch Size Experiment.** Training loss, Validation loss and Validation Dice scores for patch sizes of 128, 160 and 192

**6.4. Assessment of the proposed method on a TS's test set subject**

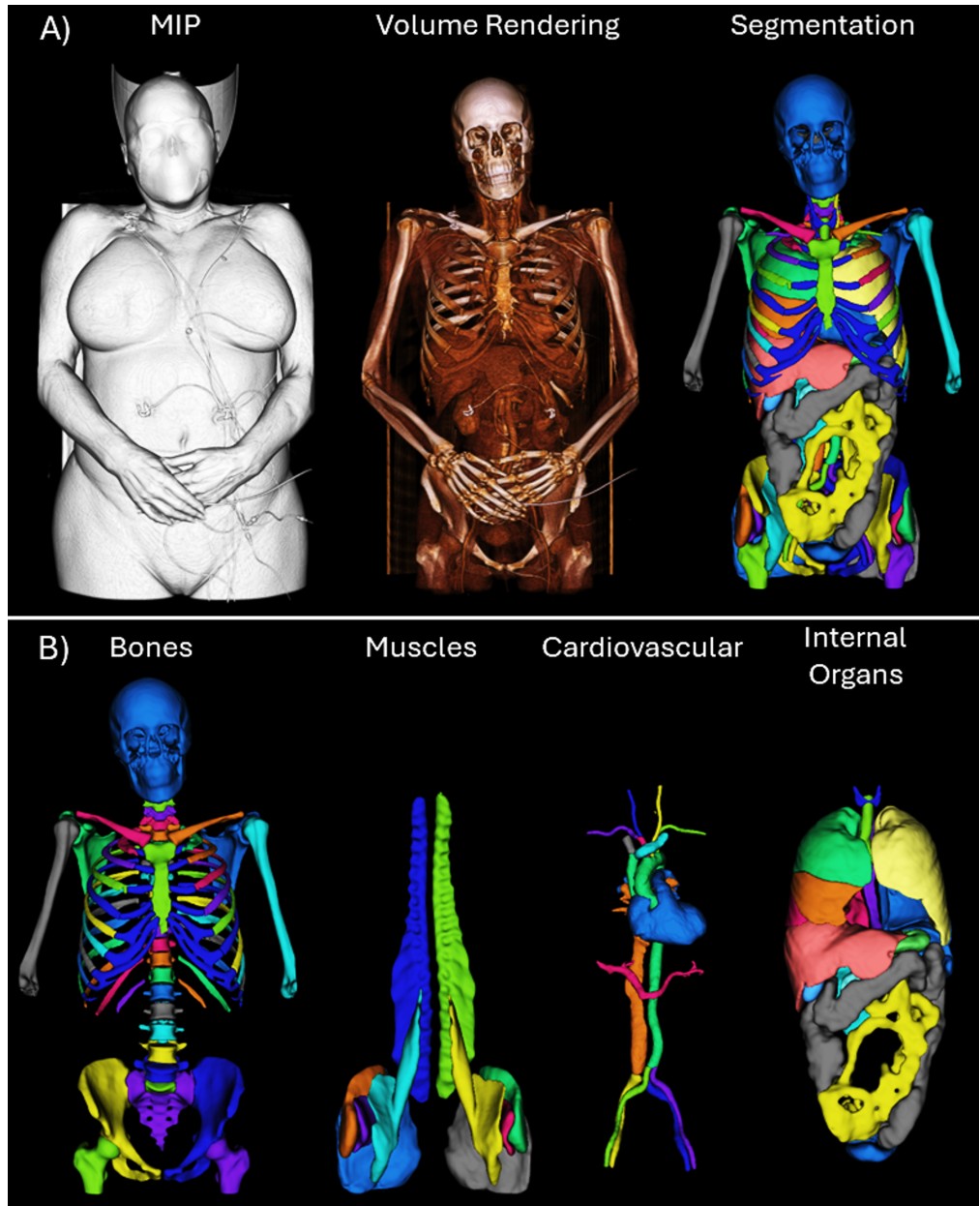

Figure 8: **3DSlicer Visualization.** A) Test subject from the TS Dataset using a 3D CT Maximum Intensity Projection (MIP) rendering, CT chest rendering and the segmentation obtained via the proposed method. B) Bone, Muscle, Cardiovascular and Internal Organ labels are shown separately for better visualization.

