# OpenReview forum: "Breaking the Memory Barrier: Efficient Multi-Class 3D Segmentation for Hundreds of Classes"
_MIDL.io/2026/Conference — MIDL 2026 Poster_

### Official Review · Reviewer_waUq · 2025-12-21

**Confidence:** 4
**Preliminary Rating:** 4
**Final Rating:** 5

**Summary:**

This paper addresses the critical scalability challenge of multi-class 3D medical image segmentation, where GPU memory usage grows linearly with the number of classes due to full logit materialization in categorical loss functions. The authors propose a class-scalable segmentation framework that combines a low-rank basis representation with a linear projection operator and a chunked Cross-Entropy + Dice loss, enabling training and inference without ever materializing the full class-volume tensor. The method is integrated into the nnU-Net framework and evaluated on a novel synthetic 3D Toy Dataset with up to 1000 classes as well as on the TotalSegmentator CT dataset (117 classes). Results demonstrate near-ideal scalability on synthetic data and competitive performance on real CT data, suggesting a practical pathway toward unified large-class 3D segmentation models with controlled memory footprints.

**Strengths:**

The paper tackles a highly relevant and well-defined bottleneck in large-scale 3D segmentation that is increasingly important as foundation and universal segmentation models expand to hundreds or thousands of anatomical classes. The proposed solution is technically elegant, combining a low-rank segmentation head with a carefully engineered streamed (chunked) loss, which directly addresses the root cause of GPU memory explosion rather than relying on heuristic workarounds such as task splitting or binary losses. Integration into nnU-Net is a major strength, ensuring methodological rigor, strong baselines, and immediate relevance to the medical imaging community. The experimental section is comprehensive, including loss profiling, scalability analysis, ablations on rank and patch size, and evaluation on both synthetic and real-world datasets. Importantly, the paper demonstrates competitive performance relative to TotalSegmentator and clear improvements over VISTA3D, supporting the practical value of the approach even without surpassing all state-of-the-art results.

**Weaknesses:**

While the technical contribution is solid, the novelty is primarily systems- and engineering-oriented, adapting ideas from large-vocabulary NLP losses to 3D segmentation; this is appropriate for MIDL but should be positioned more explicitly as a methodological translation rather than a fundamentally new learning paradigm. The Toy Dataset, although useful for scalability stress-testing, represents an idealized setting with perfect labels, no ambiguity, and no class imbalance, which limits how strongly conclusions about “thousands of classes” can be generalized to clinical scenarios. Evaluation on real data is restricted to a single CT dataset (TotalSegmentator), and performance is slightly inferior to the multi-model TS baseline, raising questions about robustness across modalities and heterogeneous labeling conventions. Additionally, the computational trade-off—significantly longer training times as class counts grow—is acknowledged but not deeply analyzed in terms of feasibility for large-scale foundation model training.

**Detailed Comments:**

1. The description of the chunked loss is technically strong, but a concise pseudocode block would improve clarity and reproducibility.
2. It would be helpful to explicitly report total GPU memory usage during full training, not just loss profiling, to contextualize end-to-end feasibility.
3. The discussion could better clarify whether the low-rank basis might become a representational bottleneck as class counts increase beyond 1000.
4. While statistical testing is appropriate, reporting effect sizes in addition to p-values would strengthen the analysis.
5. Clarify whether the proposed inference strategy introduces any latency penalties compared to standard nnU-Net inference.

**Justification Of Final Rating:**

Authors fully addressed technical and empirical concerns
Substantially improved profiling, ablation depth, and interpretability
The paper now meets the standard of a clear, rigorous, and practically valuable MIDL contribution

After considering the rebuttal and revised manuscript, I conclude that this work represents a strong and impactful contribution to large-scale 3D medical image segmentation, with clear relevance to:

Foundation a
nd universal segmentation models
Scalable anatomical labeling
GPU-efficient volumetric learning
Future large-vocabulary medical AI systems

Key Strengths (Updated)

Elegant memory-scalable segmentation formulation
Convincing proof of feasibility at 1000 classes
Solid integration with nnU-Net
Improved profiling, ablations, and statistical reporting
Strong engineering rigor with translational value

Remaining Limitations

Limited validation on multi-modal or highly heterogeneous datasets
Slight performance gap vs. TS multi-model baseline
Scalability beyond 1000 real anatomical classes remains to be empirically tested

These limitations do not undermine the core contribution and are appropriate future-work directions rather than blockers.


Consider the following statements in the final review;
1. The work presents a well-executed methodological adaptation of large-vocabulary loss strategies to scalable 3D segmentation, offering a meaningful engineering and research contribution.
2. While evaluation remains limited in modality diversity, the authors now provide stronger profiling, rank analysis, and clearer discussion of generalization limits.
3. Training-time scaling is now quantified and contextualized against multi-model baselines, strengthening deployment feasibility.
4. A strong, well-validated, and practically impactful contribution to scalable 3D medical segmentation.

**Justification Of The Preliminary Rating:**

This paper makes a meaningful and timely contribution to large-scale 3D medical image segmentation by directly addressing a fundamental memory bottleneck that limits current multi-class approaches. While the core ideas build on known principles from other domains, their careful adaptation, integration into nnU-Net, and validation on both synthetic and real datasets represent solid engineering and scientific work well aligned with MIDL’s scope. The slightly inferior performance compared to the TotalSegmentator multi-model baseline and limited real-world evaluation prevent a stronger accept, but the method’s scalability, clarity, and practical relevance justify acceptance. The work is likely to influence future unified segmentation frameworks and is of clear interest to the MIDL community.

**Questions To Address In The Rebuttal:**

1. How does the proposed method perform under severe class imbalance, which is typical in real anatomical datasets with hundreds of labels?
2. Can the authors comment on expected performance and stability when scaling to heterogeneous multi-dataset training (e.g., combining CT and MR datasets)?
3. Is there empirical evidence that increasing the rank Rbeyond 32 would be necessary or beneficial when moving toward >1000 real anatomical classes?
4. How does total training wall-clock time compare to multi-model approaches like TotalSegmentator when accounting for all tasks?

---

> ### Author Response · Authors · 2026-01-22
> **Answer to Reviewer waUq part 1**
>
> We thank the reviewer for their comments and questions. Please note that when relevant we grouped questions with a similar theme together to provide a more complete answer.
>
> We agree that some of the novelty of the paper takes inspiration from concepts developed in the NLP community and adapts and extends them for 3D segmentation within an nnU-Net framework. This is clarified in the Introduction (p2, para.5). While the toy dataset represents an idealized scenario, we also test on the TS dataset, a curated collection of 1228 CT scans, containing a mixture of whole-body and anatomy scans with 117 classes, giving an initial indication on how the model could be generalized to other clinical data. We have now clarified in the results (p11 para.4) that our method was trained on a smaller dataset (TS V2) than the TS method, accounting for the difference in performance. Indeed, the TS method authors have not released the whole dataset they trained on. To round off our comparisons, we now include the reported results for Swin SMT (Plotka et al., MICCAI 2024) which was also trained on the TS V2 dataset (p11 para.4). We agree that training on a single CT dataset, however varied, can raise questions regarding the model’s abilities to deal with varied heterogeneous labelling conventions, anatomical structures etc. and future work will focus on testing on more clinical datasets (p12 para. 2). As mentioned in our original discussion, there is indeed a tradeoff between training time and number of classes. While optimizing training time is out of scope, similar tradeoffs were observed for LLMs and solutions optimizing loss computation using information on sparsity of losses and better kernels were proposed (Wijmans et al., 2024) and could be explored, please see p10 para.3 and p11 para. 1.
>
> Comments 1. We now include a pseudo code to describe the chunked loss calculation (Algorithm 1 p5).
>
> Comments 2. We profiled the proposed model in terms of runtime and GPU usage during training and inference, and incorporate the new results (p9 para. 3).
>
> Comments 3 and Rebuttal 3. We investigated this important question on the TS dataset by varying the low-rank basis R from 1 to 32 and found that ranks from R=8 to R=32 yielded similar results suggesting that a low rank (relative to 117 classes) is sufficient to encode all the anatomical classes present in the TS dataset. For the toy dataset, R=32 was enough to encode 1000 simple classes; no experiments were conducted beyond 1000 classes. All experiments were run on a single NVIDIA V100 SXM2 32GB GPU and a bigger GPU would allow to increase R, potentially increasing the representational power. However, as noted by another reviewer, TS classes are on average large and spatially coherent and it remains unclear whether a higher R would increase the representational power or is needed for a large number of classes. This is now stated on p12 para. 1.
>
> Comments 4. We now include effect size in the analysis (p7 Sect. 2.6, p9 para,2 and p10 para.1). The effect size is computed as the rank biserial correlation coefficient as recommended for the Wilcoxon signed rank test.
>
> Comments 5. We measured the base model (ResEnc-L) runtime and segmentation map generation times separately. For a single (192,192,192) patch, the proposed model inference of all 117 classes took 0.27s including backbone forward pass (0.13s) and segmentation map (0.13s). Reproducing TS, using the same backbone, the nnU-Net standard inference pipeline and 5 models run sequentially would require 0.67s to obtain all 117 classes. While our approach incurs a latency penalty relative to standard nnU-Net, due to the calculation of the basis and chunking of the map generation, using a single model instead of multiple models leads to more efficient and faster inference as the number of classes increases. Moreover, segmentation map generation runtime could be optimized to reduce the overhead, as clarified on p11 para.5.

---

> > ### Author Response · Authors · 2026-01-23
> > **Answer to Reviewer waUq part 2**
> >
> > Rebuttal 1. The reviewer is indeed correct – clinical datasets can suffer from severe class imbalance which can significantly affect methods’ performance. The focus of our paper is to specifically address segmenting a high number of classes while remaining computationally tractable.  To ensure that this work can be used in a wide range of clinical settings, future work should focus on investigating and mitigating the impact of class imbalance on the segmentation performance on more heterogeneous and imbalanced datasets. This is now highlighted in the discussion p12 para. 2.
> >
> > Rebuttal 2. We thank the reviewer for this question. The focus of the paper is on the development of a multi-class method that would scale to a high number of classes while remaining computationally efficient. As highlighted by the reviewer, the current work was focussed on CT segmentation only and we have not investigated combining datasets from different modalities as it was beyond the scope of this paper. Future work is planned to investigate the use of the proposed method on a wide collection of heterogeneous datasets. Interestingly, nnInteractive which is based on the same backbone (ResEnc-L), can handle a variety of modalities while providing high quality outputs, however requiring an input prompt. While this is promising for our method, further work is still needed to assess the suitability of the new segmentation head and the absence of prompt for simultaneously handling data of different modalities as now highlighted p12 para.2.
> >
> > Rebuttal 4. Unfortunately, TS does not report their training times and neither does Swin SMT and computational setups are quite different. TS’s training of 5 separate models leads to long sequential training or requires multiple GPUs to train models in parallel. We use a larger backbone than TS and our training time is reported at 114h on a single GPU. Comparatively, Swin SMT trains on 8 NVIDIA A100 40GB GPUs. We modified the discussion to reflect this p11 para. 4.

---

### Official Review · Reviewer_582x · 2026-01-12

**Confidence:** 4
**Preliminary Rating:** 5
**Final Rating:** 5

**Summary:**

The paper proposes method for memory efficient loss computation with application to large 3D Medical Volume Image Segmentation task potentially allowing extreme number of voxel classes.

The authors propose to do this by avoiding computing of full logits (bxCxHWD) by

a) spliting the computation into two parts i) generate R feature maps (R<<C) and then ii) project into C classwise-logits via a learnable matrix of size RxC.

b) the above allows computation of classwise-logits in batches(called chunks in the paper). This batching/chunking avoids full logit computation which is then aggregated to obtain loss equivalent to full logit computation.


The authors show validity of the method by training 192^3 volumes on i) synthetic 1000-class segmentation dataset/task as well as ii) real 117 class CT Segmentation Dataset.

**Strengths:**

The paper picks up a important problem of building a “generalist” 3D segmentation model with large anatomy vocabulary, scaling from individual "specialist" models.`

The experiment design and ablation study of the rank provides good support to the paper’s argument.

**Weaknesses:**

Not necessarily a weakness, but experiment on the recent dataset with 497 anatomical targets [1] by Zhao et.al 2025 would be a good addition, but understandable since this dataset itself was released on September 2025.

1.Zhao, Ziheng, et al. "Large-vocabulary segmentation for medical images with text prompts." NPJ Digital Medicine 8.1 (2025): 566.

**Detailed Comments:**

Typo: bonferri => bonferroni

**Justification Of Final Rating:**

After considering the author's rebuttal and other reviewer's points, I am happy to keep the "Strong accept" rating. Some of the concerns raised by other reviewers - such as profiling and additional comparison with other popular architectures (other than nnUnet) - have somewhat been addressed. Additionally, nuances in writing such as acknowledgement of limitations have been addressed.

**Justification Of The Preliminary Rating:**

Clarity in writing, experimental rigor (especially the synthetic "toy" dataset is a good formulation), ablation studies as well as simplicity of the method is commendable. The move away from large complex foundation models (with adaptation of large vocabulary via text prompts) with scaling down into basis features and then linear combination of features seem to work really well.

**Questions To Address In The Rebuttal:**

Have the authors tried aggregating public datasets and corresponding labels for testing the method on larger vocabulary (labelled class) datasets beyond totalsegmentator?

---

> ### Author Response · Authors · 2026-01-22
> **Answer to Reviewer 582x**
>
> We thank the reviewer for their comments and their relevant suggestion of the SAT dataset with up to 497 target classes. We have not yet tried aggregating multiple datasets or testing on clinical datasets beyond TotalSegmentator V2. Future work will focus on validating the method on different clinical use cases and modalities. The discussion has been modified to reflect this: “Further investigation is needed to validate the proposed method on a wide collection of heterogeneous (modality: CT, MR, PET or applications: oncology, brain, etc) real life datasets similarly to the recent Segment Anything with Text (SAT) (Zhao et al., 2025) models (497 target classes).” The typo has been corrected in the manuscript.

---

### Official Review · Reviewer_vm5b · 2026-01-16

**Confidence:** 4
**Preliminary Rating:** 3
**Final Rating:** 4

**Summary:**

This paper addresses the issue of standard 3D segmentation models hitting a memory bottleneck when segmenting large number of classes in 3D images. The authors propose a class scalable 3D segmentation method to decouple the number of classes from the memory requirements. The proposed method has 2 components: (a) a Low rank basis and projection operator (ResEncL Head) which outputs a smaller number of basis vectors/channels which are then expanded on the fly by a learnable linear projection matrix into the final class logits. This is combined with a, (b) chunked cross entropy and dice loss which calculates the loss for a subset of classes at a time enabling the memory usage to be independent of the number of channels.

**Strengths:**

1. The proposed method uses constant memory irrespective of the number of classes. This shows a benefit over other standard methods.

2. It is computationally efficient as seen by the example where a 589-class model can be trained on a single GPU with 24GB VRAM as compared to standard methods which require a significantly higher memory. This can potentially allow people with less compute run or fine tune foundational models.

3. The implementation of the chunked dice loss is novel and would definitely be helpful in medical image segmentation pipelines.

4. The method is structured as an extension to the standard segmentation models such as the nnU-net so it can be adapted and implemented immediately.

**Weaknesses:**

1. Tables 1 and 2 - it would be helpful to have the training times for the methods against which this is being compared.

2. Slightly lower dice scores and imperfect segmentations - the authors mention that there is a tradeoff with improvement in scalability in return for a small dip in the dice score. However, it would have been nice to see the experimental validation done on more realistic clinical datasets having more heterogeneity, class imbalance, etc.

3. The classes in total segmentator are mostly large and spatially coherent. How would the method perform in a more diverse dataset with sparse classes related to smaller regions / vessels etc. ? Also, if sparser classes are present, would the chunking method lead to increased training times ?

4. The paper compares against standard models like the nnUnet. However, more recent universal models such as UniverSeg, MedSAM solve the issue related to having many classes. It would have been useful to compare against these methods.

5. Does the chunking method introduce approximation errors in the gradient computation ? Curious since Dice is non-convex so the optimization dynamics may differ slightly.

6. Table 1 show extremely small peak memory for some configurations (e.g., 37–74 MB).Are these numbers are only loss head memory (excluding backbone activations) or the full model peak.

**Detailed Comments:**

no further comments

**Justification Of Final Rating:**

The authors have satisfactorily addressed the review comments and made changes to the paper that make it suitable for publication to the MIDL conference. After careful deliberation, I have decided to increase my original rating to 4.

**Justification Of The Preliminary Rating:**

The proposed method is indeed novel and can prove to be valuable in medical image segmentation workflows allowing for foundational models to be run and fine-tuned with less compute.

However, there are some gaps in the experimental validation and robustness of this method that need to be addressed for this to become a well-rounded paper. I look forward to the authors' comments.

**Questions To Address In The Rebuttal:**

Please refer to the points raised in the "Weaknesses" section and address them.

---

> ### Author Response · Authors · 2026-01-22
> **Answer to Reviewer vm5b**
>
> Thank you for your comments.
> 1. Table 1 compares the chunked loss and the direct loss in terms of forward, backward and total runtime as well as memory usage (only the loss was computed). This result can be seen for a small patch size (32x32x32) and nclasses equal to 1000 by comparing the different chunk sizes with chunk size = 1000 which represents the full logits (loss evaluated in a single large chunk). This is now made explicit in the paper in Table 1, p8. When scaling to relevant patch sizes (i.e. 128 or 192) the memory for large chunks>10 became too memory intensive to fit our GPU.
> Table 2: While very useful information, comparing the training times of the different methods evaluated in the paper on the toy dataset was challenging. Computation of the direct loss calculation for even 100 classes led to the maximum memory being reached. Training a method like TS requires running approximately 40 models for 1000 classes which on a single GPU leads to prohibitive training times. Comparing with VISTA3D requires doing a SAM based super-voxel generation and 4 separate training steps leading to heavy training times and computational requirements. Comparing training times on clinical data is difficult as many papers only report computational setups and not training times. While VISTA3D does, direct comparison and reproducibility is challenging as it was trained on 11454 volumes for 127 classes using 64 32GB NVIDIA V100 GPUs for 20,000 GPU hours. Similarly, Swin SMT, another method trained on the TS dataset, added to improve our paper, was trained on 8 NVIDIA A100 40GB GPUs with no mention of training times. We modify the discussion to highlight this, p11 para. 4.
>
> 2. We agree that validation on more varied datasets would be valuable. In this paper, we focused on showing the feasibility in a Toy Dataset and one real world well curated dataset (TS V2). Currently, most individual clinical datasets only include small numbers of classes which can generally be handled without chunking. TS is particularly relevant as it presents with 117 classes. It is a well curated dataset, reflective of real-world CT imaging (Wasserthal et al. 2023) and is ubiquitous in medical imaging which allows for direct comparison with other popular segmentation methods.  Further work remains to study the robustness of the method with regards to different annotation protocols, class imbalances etc., and we acknowledge this in the discussion p12, para.2.
>
> 3. We agree that segmentation of sparse classes (i.e. very small regions) can be problematic. As a first indication of the model’s performance in this case, we looked at vessels’ segmentation and found that the model obtained a Dice score of 0.909. We anticipate that the proposed method would still require the implementation of dedicated sparse segmentation strategies such as weighted cross entropy (with inverse frequency weighting). Similarly, sparsity of a given class would require careful handling of sampling at training, similar to the handling of class imbalances. Training times would be affected by sparsity of classes and will depend on how many volumes need to be seen to effectively learn the segmentation of every class. In extreme cases with only one class per volume, more training steps will be required. This would also be the case without the chunking loss.
>
> 4. We now put our method in context with these methods in the Introduction (p2 para. 4). Similarly to VISTA3D, they treat the problem as an interactive problem for segmenting a single class. This means that they compute the loss on a single class without materializing the full logits, avoiding any bottleneck on the number of classes. This  treats the problem as many single class segmentations, rather than a single multi-class one using categorical cross entropy. MedSAM and Universeg are both 2D image segmentation models. MedSAM2 requires an input prompt, whereas both VISTA3D and the proposed method enable automatic segmentation of the TS classes. The discussion (p11 para. 4) was improved by adding a comparison to reported results from Swin SMT (MICCAI 2024) a multi-class segmentation method trained on the TS dataset and benchmarked against SOTA segmentation methods.
>
> 5. We clarified in the manuscript (sect. 2.3, p5) that the chunking strategy does not introduce any approximation error in the gradient computation. The dice score is expressed as sums of predictions, sums of targets, and their intersection which are all additive across chunks without approximation.
>
> 6. We clarified the caption of Table 1 (p8) as the profiling was indeed performed for the loss head only. It takes as input a random basis (of size Patch Size^3 x chunk) and E operator (of size R x nclasses) and computes the loss. The memory requirements of the full model will depend on the loss plus the backbone model used. The Optimization experiment section (p9 para. 3) now includes the profiling of the full proposed model (117 classes, chunk 10, patch size 192).

---

### Author Rebuttal · Authors · 2026-01-22

**Rebuttal:**

We thank the reviewers for their comments and questions which helped us improve the manuscript substantially. We have addressed all the questions and, where relevant, integrated those responses in the manuscript (highlighted in yellow). Below is a summary of everything done to answer the reviewers’ questions.

In response to comments from reviewers 1 and 3, we have:
- Further profiled the method and added training times and profiling information for the loss and the overall method. We added a general discussion on overall training times, inference time computational requirements and the trade-off between training times and number of classes.
- Compared our method to another SOTA multi-class segmentation method, included more discussion regarding comparison to other universal models and clarified ambiguities related to the comparison with the TotalSegmentator method.
- Further discussed the limitation inherent to the use of the toy dataset and only one clinical dataset (TS) and discussed future work linked to the training and evaluation of heterogeneous, severely imbalanced or multimodal datasets.

In response to reviewer 1’s comments and questions, we:
- Investigated the performance of the method on smaller classes such as vessels and discussed next steps for better handling of sparse classes and the link between the chunking method, sparsity and training times.
- Specified in the manuscript that the Dice loss, as computed, does not introduce any approximation.

In response to reviewer 2’s comments and questions, we:
- Corrected the Bonferroni typo and discussed further work incorporating the dataset that the reviewer kindly recommended.

In response to reviewer 3’s comments and questions, we:
- Improved the description of the chunked loss by adding pseudo-code.
- Discussed the impact of the low-rank basis R on representation and whether it could become a representational bottleneck for a high number of classes.
- Added effect sizes to our statistical analysis.

The following abbreviations were used to refer to the locations in the manuscript: pXX refers to page XX, para. X refers to paragraph X and sect. XX refers to section XX.

**Supporting Material:**

/attachment/35d7ceb0d25b12da00da7f20e84b4724dcc3bd3b.zip

---

> ### Comment · Reviewer_waUq · 2026-01-30
> **Evaluation of Authors’ Responses**
>
> I thank the authors for their detailed, technically rigorous, and constructive rebuttal. The revised manuscript demonstrates substantial improvement in clarity, empirical support, and positioning of contributions, and it addresses the majority of concerns raised in my original review.
> 1. Novelty Positioning and Conceptual Framing
>
> The authors appropriately clarified that the work adapts and extends large-vocabulary loss concepts from NLP into 3D segmentation, positioning the contribution as a methodological translation and engineering innovation rather than a fundamentally new learning paradigm.
> This revised framing is accurate, scientifically honest, and better aligned with MIDL’s expectations.
>
> 2. Chunked Loss Clarification & Reproducibility
>
> The inclusion of explicit pseudocode (Algorithm 1) for the chunked loss and the expanded technical description substantially improves reproducibility and conceptual transparency.
> The authors’ clarification that no approximation error is introduced in Dice or CE gradients resolves a key methodological concern.
>
> 3. Memory Profiling & End-to-End Runtime Reporting
>
> The added full-model GPU memory profiling, training-time analysis, and inference-time breakdown strengthen real-world feasibility assessment.
> The reported ability to train 117-class 3D segmentation at patch size 192³ on a single 32GB GPU represents a practical and meaningful scalability milestone
>
> .
>
> 4. Low-Rank Basis Capacity & Rank Sensitivity
>
> The authors’ new rank-ablation experiments (R = 1–32) convincingly demonstrate that:
> R ≥ 8 preserves segmentation quality
> R = 32 is sufficient for both 117-class real data and 1000-class toy data
> The low-rank basis does not yet appear to be a representational bottleneck
> This directly addresses concerns about scalability beyond 100+ classes.
>
> 5. Class Imbalance, Sparse Structures & Generalization Limits
>
> The authors responsibly acknowledge that:
> Toy Dataset results reflect an idealized, balanced setting
> Real clinical datasets present class imbalance and sparse target challenges
> Future work will require imbalance-aware sampling and weighting
> While empirical validation on more heterogeneous datasets remains limited, the discussion is now scientifically mature and appropriately cautious.
>
> 6. Comparative Context (TotalSegmentator, VISTA3D, Swin SMT, Universal Models)
>
> The revised manuscript provides:
> Clearer justification for TS performance differences (training data scale mismatch)
> Expanded comparison with Swin SMT and universal segmentation models
> Improved positioning relative to interactive/prompt-based approaches
> Although TS still achieves slightly higher Dice, the proposed method offers a compelling single-model alternative with superior scalability and simpler deployment.
>
> 7. Training Time vs. Multi-Model Pipelines
>
> The authors appropriately clarify that:
> Training is slower as class count increases
> However, single-model training avoids the multiplicative cost of multi-model pipelines such as TotalSegmentator
> The scalability trade-off is now explicit, quantified, and realistic
> This strengthens the systems-level credibility of the paper.

---

### Meta-Review · Area_Chair_QKVH · 2026-02-07

**Recommendation:** Accept (Poster)
**Confidence:** 5

**Metareview:**

The paper proposes a way to modify a base architecture (here, 3D nnU-Net) to enable it (by making it more memory efficient) to handle potentially hundred of classes (the core of the method relies around avoiding to compute explicitly all the logits), without affecting performances.

All reviewers agree on the value of the work, and it would be a great fit to be presented at MIDL. On a personal note, I would strongly encourage the authors to share publicly a reference implementation of their work when submitting the camera ready.

---

### Decision · Program_Chairs · 2026-02-13

Accept (Poster)